# Defining the Basal and Immunomodulatory Mediator-Induced Phosphoprotein Signature in Pediatric B Cell Acute Lymphoblastic Leukemia (B-ALL) Diagnostic Samples

**DOI:** 10.3390/ijms241813937

**Published:** 2023-09-11

**Authors:** Aaruni Khanolkar, Guorong Liu, Bridget M. Simpson Schneider

**Affiliations:** 1Department of Pathology, Ann and Robert H. Lurie Children’s Hospital of Chicago, Chicago, IL 60611, USA; 2Department of Pathology, Northwestern University, Chicago, IL 60611, USA

**Keywords:** pediatric B cell acute lymphoblastic leukemia (B-ALL), signaling, phosphorylation, signal transducer and activator of transcription (STAT), immunomodulatory mediators, cytokines, CD40 ligand (CD40L), receptors

## Abstract

It is theorized that dysregulated immune responses to infectious insults contribute to the development of pediatric B-ALL. In this context, our understanding of the immunomodulatory-mediator-induced signaling responses of leukemic blasts in pediatric B-ALL diagnostic samples is rather limited. Hence, in this study, we defined the signaling landscape of leukemic blasts, as well as normal mature B cells and T cells residing in diagnostic samples from 63 pediatric B-ALL patients. These samples were interrogated with a range of immunomodulatory-mediators within 24 h of collection, and phosflow analyses of downstream proximal signaling nodes were performed. Our data reveal evidence of basal hyperphosphorylation across a broad swath of these signaling nodes in leukemic blasts in contrast to normal mature B cells and T cells in the same sample. We also detected similarities in the phosphoprotein signature between blasts and mature B cells in response to IFNγ and IL-2 treatment, but significant divergence in the phosphoprotein signature was observed between blasts and mature B cells in response to IL-4, IL-7, IL-10, IL-21 and CD40 ligand treatment. Our results demonstrate the existence of both symmetry and asymmetry in the phosphoprotein signature between leukemic and non-leukemic cells in pediatric B-ALL diagnostic samples.

## 1. Introduction

Our awareness of the association between inflammation and cancer stretches back over one hundred and fifty years when Rudolf Virchow made his astute observation documenting the presence of leukocytes in cancerous tissues [1,2,3,4,5,6,7]. However, only 15–20% of malignancies are associated with pre-existing localized inflammation, and inflammation impacts critical steps in the neoplastic process, such as tumor growth, metastatic spread and the outcome of therapeutic interventions [2,3,4,8,9,10,11,12,13,14]. Multiple lines of evidence point to the fact that the tumor micro-environment (TME) tends to be replete in inflammatory mediators, such as cytokines and chemokines [2]. In addition, consistent with this evidence, clinical data also reveal that both non-specific as well as targeted inhibition of inflammatory signals can attenuate the risk for developing certain types of cancers [15,16,17]. Even malignancies that lack microbial drivers experience inflammatory stress induced by oncogenic transformation, metabolic alterations, cell death and hypoxia [4,8,18]. Indeed, genetic lesions that disrupt the tumor suppressor function can trigger DNA damage-induced inflammatory responses, and conversely, the activation of oncogenes can augment the expression of cytokines and chemokines that can drive tumor progression and/or simultaneously inhibit anti-tumor immune responses [8,19,20,21,22,23,24].

B cell acute lymphoblastic leukemia (B-ALL) is the most frequent cancer in children [25]. In fact, pre-leukemic clones harboring the *ETV6-RUNX1* gene fusion are observed in almost 5% of healthy newborns, but frank leukemic transformation occurs in only a small fraction (~0.2%) of children that carry this genetic aberration [26,27,28]. Prevailing wisdom suggests that dysregulated immune responses to childhood infectious insults contribute to the accumulation of additional genetic hits that ultimately drive the development of clinically apparent B-ALL [26]. Thus, B-ALL represents an aberrant outgrowth of a primitive B cell clone that is endowed with a selective growth advantage aided and abetted by a varied set of genomic alterations involving coding DNA sequences as well as non-coding genetic elements [25]. This underlying genetic heterogeneity underpins the spectrum of clinical phenotypes and therapy-related outcomes observed in pediatric B-ALL patients. Although the conventional five-year survival rate in pediatric B-ALL patients exceeds 85%, disease relapse is associated with a high mortality rate [25,29]. Collectively, available data suggest that a combination of at least two factors can influence the development of pediatric B-ALL. These factors include, first, an underlying genetic predisposition characterized by the aforementioned variably penetrant, somatic or germline genetic variants; and second, infectious insults that elicit atypical responses, including a disordered cytokine profile, from an untrained immune system [26].

Beginning with Kinlen’s “population-mixing” theory outlined in 1988, in the ensuing years, at least five other theoretical models have been proposed, aimed at enhancing our understanding of the elements that might promote the onset of pediatric B-ALL [26,30]. These alternate models include Greaves’ delayed immune exposure theory; the infective lymphoid recovery hypothesis; Smith’s theory, which entails the vertical transmission of infections from the mother to the fetus; Schmiegelow’s adrenal hypothesis model; and an emerging model based on the concept of “trained immunity” [31,32,33,34,35]. A common thread that runs through these theoretical models is the putative link between childhood infections and perturbations in immunomodulatory cytokines elaborated by the host’s immune system that is still “learning the ropes”. Previous investigations have revealed perturbations in cytokine profiles of neonates that later go on to develop B-ALL in comparison to children that remain healthy [26,36,37,38]. It has also been demonstrated that allelic polymorphisms that affect the production of key effector cytokines, such as IFNγ, can influence the timing of the onset of B-ALL in children, and a subset of pediatric B-ALL patients that relapse and display IL-15 receptor α (*IL15Rα*) gene expression levels in blasts that are above the median level experience better subsequent event-free and overall survival [39,40]. Altogether, these lines of scientific evidence further support the notion that either direct cross-talk between precursor B cells and T cells, or perhaps bystander responses to soluble mediators such as cytokines, elicited during childhood immune responses could potentially support the malignant outgrowth of precursor B cells that harbor deleterious genetic variants.

Cell signaling undergirds almost every aspect of human biology, and signaling associated with immunomodulatory mediators, such as cytokines and co-stimulatory molecules, plays a pivotal role in the lineage commitment, maturation, differentiation and functional polarization of many immune cell types [41,42,43,44,45,46,47]. Very little is known about immunomodulatory-mediator-induced signaling responses in B-ALL blasts and/or co-existing non-malignant lymphocytes in pediatric B-ALL samples, and our knowledge of the signaling potential and properties of the leukemic blasts existent in diagnostic samples of pediatric B-ALL patients is incomplete [40]. Therefore, gaining a deeper understanding of the signaling properties of malignant immune cells and their normal counterparts in diagnostic samples of pediatric B-ALL subjects might offer clues that could potentially be leveraged in the future as actionable biomarkers that allow for more precise disease classification as well as to identify novel therapeutic targets. Because the behavior of leukemic B-ALL blasts is so clearly aberrant compared to non-leukemic B cells, we hypothesized that the immunomodulatory-mediator-driven phosphoprotein signature in the blasts completely diverges from that of the non-leukemic B cells residing in the diagnostic B-ALL samples. To test this hypothesis, we assembled a data set defining the immunomodulatory-mediator-induced signaling landscape of leukemic blasts, as well as normal mature B cells and T cells residing in diagnostic samples from 63 pediatric B-ALL patients. We utilized phosflow to evaluate the phosphorylation status of proximal signaling nodes following the treatment of diagnostic B-ALL samples with immunomodulatory mediators that are often generated and/or upregulated in the setting of pathogen-induced immune responses. Specifically, recombinant human (rh) IFNγ, IL-2, IL-4, IL-7, IL-10, IL-21 and trimerized CD40 ligand (CD40L) were used as signaling inputs to define this phosphoprotein signature. These signaling studies were complemented by a detailed evaluation of the surface expression patterns for the following cytokine receptors: IFNγR1, IFNγR2, CD25 (IL-2Rα), CD122 (IL-2Rβ), CD124 (IL-4Rα), CD127 (IL-7Rα), CD132 (common γ chain), IL10Rα, IL-10Rβ, CD360 (IL-21R) and CD40 in a subset of the patients. To the best of our knowledge, this study represents one of the few (if not the only) report(s) of a comprehensive assessment of basal and immunomodulatory-mediator-induced phosphoprotein signatures in diagnostic samples of pediatric B-ALL subjects. Furthermore, the identification of significantly elevated basal phosphorylation levels of STAT(s)-1, 5, 6 and p65-NFκB in the diagnostic samples of patients that experienced disease-related mortality is a novel observation, as it underscores the potential prognostic value of assessing phosphorylation signals at the time of diagnosis.

## 2. Results

### 2.1. Basal Hyperphosphorylation in Leukemic Blasts

Basal hyperphosphorylation refers to the existence of an elevated background phosphorylation state of a signaling node, detected directly ex vivo in the absence of any exogenous treatment with stimulants such as immunomodulatory mediators. The detection of basal hyperphosphorylation indirectly suggests the in vivo presence of low-level stimuli that could potentially contribute to persistent low-grade inflammation [48]. As the neoplastic process is intimately associated with inflammation, the assessment of basal phosphorylation signals can provide critical early clues relating to underlying low-grade persistent inflammatory states [1,2,3,4,5,6,7,9,10,11,12,13,14]. We examined and compared the basal phosphorylation states of leukemic blasts and non-leukemic mature B cells and T cells residing in diagnostic B-ALL samples obtained from sixty-three patients with a confirmed diagnosis of pediatric B-ALL leukemia (Figure 1; for the gating strategy, please refer to Appendix A). The signaling nodes evaluated for this assessment included the following Signal Transducer and Activator of Transcription (STAT) molecules: STAT1, STAT3, STAT5 and STAT6, and we also assessed the phosphorylation of p65-NFκB (Figure 1). The basal phosphorylation state of the leukemic blasts was significantly elevated compared to the neighboring mature B cells in the same sample for STAT1, STAT3, STAT5, STAT6 and p65-NFκB (Figure 1). Four to eight samples obtained from non-leukemic donors were similarly evaluated to assess basal hyperphosphorylation among hematogones and mature B and T cells. We were not able to detect similar evidence of basal hyperphosphorylation in hematogones compared to mature B cells and CD3+ T cells that were present in non-leukemic bone marrow samples (for the gating strategy, please refer to Appendix A). We also assessed the basal phosphorylation status of the leukemic blasts against the hematogones, which revealed that leukemic blasts had significantly elevated basal phosphorylation for STAT(s)3, 5, 6 and p65-NFκB but not STAT1 (Appendix A). Overall, these data suggest that an altered response to local environmental cues is hardwired into the signaling machinery of leukemic cells.

### 2.2. Phosphoprotein Signature of Pediatric B-ALL Blasts Treated with Immunomodulatory Mediators

Immunomodulatory mediators imprint the trajectory of development and maturation, as well as the functional attributes of lymphocyte types [41,42,43,44,45,46,47]. Therefore, we undertook a comprehensive survey of the immunomodulatory-mediator-induced signaling properties of leukemic blasts and mature B and T cells present in pediatric B-ALL diagnostic samples. As part of this effort, we evaluated the phosphorylation status of proximal signaling nodes that engage with the cognate receptors for the following cytokines: IL-7, IL-2, IL-4, IL-21, IFNγ and IL-10, as well as the phosphorylation of p65-NFκB following the engagement of the CD40 receptor (Figure 2a–g) [49,50,51,52,53,54,55]. The majority of responses were clearly divergent between the leukemic blasts and mature B cells. For instance, the IL-7-induced pSTAT5 signal was absent in mature B cells but clearly detectable in leukemic blasts, whereas the IL-4, IL-21, IL-10 and CD40L-induced signaling responses were attenuated in leukemic blasts but robust in mature B cells (Figure 2a,c,d,f,g). Interestingly, the signaling responses induced by IL-2 and IFNγ displayed good symmetry between leukemic blasts and mature B cells (Figure 2b,e). Specifically, treatment with IFNγ evoked a robust pSTAT1 signal in both leukemic blasts and mature B cells, and IL-2-induced STAT5 phosphorylation was undetectable in both subsets. In a few experiments, we also examined the signaling properties of hematogones present in non-leukemic bone marrow specimens, and this allowed us to separately compare the phosphoprotein profile of hematogones with that of the leukemic blasts. This set of assessments also revealed evidence of signaling symmetry as well as asymmetry between these two populations. IFNγ-induced STAT1, IL-2-induced STAT5 and IL-10-induced STAT3 phosphorylation was comparable between the leukemic blasts and hematogones, and the IL-4, IL-7, IL-21 and CD40L-induced phosphorylation of STAT(s)6, 5, 3 and p65-NFκB displayed significantly elevated signals in leukemic blasts (Appendix A). For both leukemic and non-leukemic samples, T cells were assessed in parallel and served as crucial internal controls for the phosflow assays (Figure 2). Overall, these data indicate both the preservation as well as the loss of select signaling properties associated with leukemic blasts that diverge from the normal maturation pathway and fail to differentiate into mature B cells.

### 2.3. Cytokine Receptor Expression Profile

In an attempt to better comprehend the observed symmetry and asymmetry in the signaling profiles of leukemic blasts and mature B cells in diagnostic leukemic samples as well as hematogones in non-leukemic specimens, we undertook an evaluation of the cognate surface receptors for the immunomodulatory mediators in a separate series of experiments. Specifically, we examined the expression of IL-2Rα (CD25), IL-2Rβ (CD122), IL-4Rα (CD124), IL-7Rα (CD127), IL-10Rα (CD210), IL-10Rβ, IL-21R (CD360), CD40, IFNγR1 (CD119), IFNγR2 and the common γ chain (CD132) on the surface of the relevant cellular subsets.

#### 2.3.1. IL-7Rα (CD127) and the Common γ Chain (CD132)

As has been described in the literature, a greater frequency of leukemic blasts expressed IL-7Rα on their surfaces compared to mature B cells, and a similar trend was observed when we compared the median fluorescence intensity (MFI) of surface IL-7Rα (Figure 3a,b) [56,57,58,59,60]. This observation was consistent with our initial finding that IL-7-induced STAT5 phosphorylation was significantly elevated in leukemic blasts compared to mature B cells (Figure 2a). Hematogones also expressed higher levels of IL-7Rα compared to their mature counterparts in non-leukemic specimens, but this level was significantly lower compared to leukemic blasts (Figure 3c). The common γ chain (CD132) is the signaling subunit shared by IL-2, IL-4, IL-7, IL-9, IL-15 and IL-21 [49,50]. In order to further resolve the disparate signaling profile observed in leukemic blasts and mature B cells associated with IL-7 treatment, we also examined the surface expression of CD132. A greater frequency of leukemic blasts expressed CD132 compared to mature B cells in diagnostic B-ALL samples, and additionally, the density of CD132 surface expression was also slightly greater in the leukemic blasts (Figure 3d,e). Interestingly, we did not detect any significant difference between leukemic blasts and hematogones in non-leukemic bone marrow in terms of the frequency of these cells expressing CD132, as well as the MFI of CD132 expression (Figure 3f).

#### 2.3.2. IL-2Rα (CD25) and IL-2Rβ (CD122)

In addition to its engagement downstream of IL-7R, STAT5 is also engaged downstream of IL-2R, and stimulation with IL-2 revealed a symmetrical lack of STAT5 phosphorylation in leukemic blasts and mature B cells (Figure 2b). This symmetry was preserved in the pattern of the expression of IL-2Rα (CD25) and IL-2Rβ (CD122) between the leukemic blasts and mature B cells, as well as between hematogones and mature B cells in non-leukemic specimens (Figure 4a,b and Appendix A).

#### 2.3.3. IL-4Rα (CD124)

As mentioned above, CD132 is also paired with the IL-4 receptor [49,50]. With respect to IL-4-driven STAT6 phosphorylation, we detected a much stronger signal in the mature B cells in the leukemic samples compared to the blasts, and accordingly, we only detected IL-4Rα surface expression on mature B cells (Figure 2c and Figure 4c). This surface receptor pattern was mirrored in the hematogones in non-leukemic specimens (Appendix A).

#### 2.3.4. IL-21R (CD360)

The fourth immunomodulatory mediator in our study that also utilizes CD132 is IL-21 [49,50]. In the context of B cells, IL-21 is predominantly involved in regulating the germinal center reaction and associated humoral immune responses, as well as plasma cell differentiation [52,61,62,63]. Under the right conditions, IL-21 has also been reported to induce apoptosis of non-Hodgkin’s B lymphoma cells, and in certain instances, it can also equip B cells to express cytotoxic effector molecules such as granzyme B [64,65,66]. Our phosflow data revealed robust IL-21 signaling in mature B cells and T cells but not in leukemic blasts (Figure 2d). Consistent with this observation, only a small frequency of leukemic blasts expressed surface IL-21R compared to mature B cells (as well as T cells) (Figure 4d). A similar trend was observed in non-leukemic bone marrow specimens, where hematogones displayed the lowest proportion of cells displaying surface IL-21R expression (Appendix A).

#### 2.3.5. IFNγR1 (CD119) and IFNγR2

Notably, symmetrical yet diametrically opposite to the IL-2-induced signaling response, IFNγ-induced STAT1 phosphorylation was observed to be equally robust between leukemic blasts and mature B cells in leukemic samples (Figure 2e). When we assessed the surface expression of IFNγ receptors, there were no significant differences in the frequencies of these two cell subsets expressing IFNγR1, although a higher frequency of mature B cells expressed IFNγR2 (Figure 5a,c). The MFI for both IFNγ receptors were slightly lower in the blasts compared to the mature B cell subset (Figure 5b,d). The proportion of leukemic blasts expressing IFNγR1 was slightly higher compared to hematogones in non-leukemic specimens, but there was no such difference noted for IFNγR2 (Appendix A). Taken together, these findings suggest that the reduced levels in the surface expression patterns of IFNγR on the leukemic blasts do not significantly hinder the formation of the IFNγ-IFNγR1-IFNγR2 ternary complex that is essential and required for the signaling cascade that engages STAT1 further downstream in the IFNγ signaling axis [67,68].

#### 2.3.6. IL-10Rα (CD210) and IL-10Rβ

STAT3 phosphorylation induced by IL-10 treatment was significantly depressed in leukemic blasts compared to mature B cells (Figure 2f). Neither the blasts nor the mature B cells expressed much IL-10Rα on their surfaces. However, greater than 80% of mature B cells in leukemic samples expressed IL-10Rβ, and ~60% of leukemic blasts displayed IL-10Rβ on their surface. However, there was no significant difference in the MFI of IL-10Rβ expression between the blasts and mature B cells in the leukemic samples (Figure 6a–d). Given this, the markedly attenuated IL-10-induced pSTAT3 signal in the blasts was intriguing. Overall, this might suggest that, in the absence of IL-10Rα expression, >80% of B lineage cells need to express IL-10Rβ to transduce IL-10 signals effectively. Alternatively, the precise molecular requirements for IL-10 signaling could also differ between blasts and mature B cells. Hematogones in non-leukemic specimens closely mirrored the leukemic blasts in terms of their IL-10 receptor expression profile as well as their IL-10 signaling potential (fold change) (Appendix A).

#### 2.3.7. CD40

CD40 is a member of the tumor necrosis factor (TNF)-receptor superfamily [69]. It is expressed on several hematopoietically-derived antigen presenting cells, including mature B cells, as well as non-hematopoietic cells [69]. In our study, consistent with published data, all mature B cells expressed CD40 on their surfaces at high levels (Figure 6e,f). It has also been previously reported that B-ALL blasts express CD40, and we did observe CD40 expression on leukemic blasts in our diagnostic samples. However, both the frequency of the blasts expressing CD40 as well as the level of surface expression was significantly lower compared to mature B cells (Figure 6e,f) [70,71,72,73,74,75,76,77,78]. This finding could potentially explain the reduced phosphorylation of p65-NFκB in the leukemic blasts following treatment with trimerized CD40L (Figure 2g) [55]. Similar trends were observed between mature B cells and hematogones in non-leukemic specimens (Appendix A).

### 2.4. Phosphoprotein Signature and Overall Survival Analyses

We also examined the phosphoprotein signature in the context of overall survival (OS) (Figure 7) [79]. The OS group included study subjects that had been classified as Standard Risk (SR), High Risk (HR) and Very High Risk (VHR) per established criteria at the time of diagnosis [25,29,80], as well as the study subjects that experienced disease relapse [25,29,80]. The study subjects that succumbed to the disease were excluded from the OS group. Interestingly, basal phosphorylation levels of STAT1, STAT5, STAT6 and p65-NFκB were significantly higher in the diagnostic samples obtained from the study subjects that experienced mortality (Figure 7). No significant differences were observed when event-free survival (EFS) was evaluated [79]. The amplified basal phosphorylation signal in the study subjects that experienced mortality might suggest the likelihood of a markedly dysregulated inflammatory milieu that may or may not be directly related to their underlying disease. Furthermore, these data also highlight the potential prognostic value of measuring the basal phosphorylation signal in diagnostic samples in terms of predicting the likelihood of survival from the disease.

## 3. Discussion

In this study, we utilized phosflow to examine the phosphoprotein signature of leukemic blasts in diagnostic samples of 63 pediatric B-ALL patients that span the spectrum of established diagnostic risk classifications (SR; HR; VHR) [25,29,80]. Accumulating evidence implicates a disordered immune response to early-life microbial insults that colludes with pre-existing (in utero) genetic lesions as well as those acquired postnatally before the disease manifests as frank B-ALL [26,30,31,32,33,34,35]. Hence, we investigated the signaling responses at proximal cytosolic signaling hubs following treatment with a defined suite of immunomodulatory mediators that are commonly generated by the host immune system in response to pathogens. These phosphoprotein signals were measured in leukemic blasts and benchmarked against the same signal detected in neighboring mature (un-transformed) B cells and T cells present in the diagnostic samples. In an effort to better understand the perceived differences in the phosphoprotein signals among the blasts, mature B cells and T cells, we also conducted an assessment of cytokine receptor and CD40 surface expression patterns on these cellular subsets in a smaller group of patient samples.

We detected evidence of basal hyperphosphorylation in leukemic blasts compared to mature B cells in diagnostic B-ALL samples across all five signaling nodes, STAT(s)-1,3,5,6 and p65-NFκB, assessed in this study (Figure 1). This finding is consistent with the notion that inflammation is writ large in the tumor microenvironment (TME) [2,8,14,81]. We speculate that this persistent hyperphosphorylation is quite likely the result of a combination of ongoing low-grade cytokine-induced signaling in the TME, coupled with altered calibrations in the sensitivity thresholds for cytokine responsiveness in the leukemic blasts. The fact that neighboring untransformed mature B cells in the diagnostic samples and hematogones in non-leukemic samples do not display evidence of basal hyperphosphorylation suggests that aberrations in the signaling machinery are specifically localized to the blast population in the sample (Figure 1 and Appendix A). In future studies, we will endeavor to define the cytokine landscape existent in the B-ALL diagnostic samples in order to obtain a better handle on the specific cytokines that are likely contributing to this basal hyperphosphorylation. Given that signaling pathways for different cytokines can intersect and converge at several signaling hubs, dissecting this network and parsing out these signaling pathways will entail in-depth computational analyses that are beyond the scope of this current study. Another limitation of our study is the bulk analyses of the B-ALL blasts and mature B cells. Follow-up studies will include assessments at the single-cell level to enhance the resolution of the analyses.

In general, the amplitude of the phosphorylation signal for the leukemic blasts assessed in response to the immunomodulatory mediators used in the study was not particularly robust, with the exception of IFNγ-induced signaling (Figure 2e). The reduced signaling potential in response to IL-2, IL-4 and IL-21 may be attributable to the reduced surface expression levels of the cognate receptor subunits [IL-2Rα (CD25), IL-2Rβ (CD122), IL-4Rα (CD124) and IL-21R (CD360)] because the frequency and density of the expression of the common γ chain (CD132), the signaling subunit shared by these three cytokines, was higher in the leukemic blasts versus the mature B cells (Figure 2b–d, Figure 3d,e and Figure 4a–d). Furthermore, a significantly lower proportion of leukemic blasts expressed IL-10Rβ on their surfaces compared to mature B cells in the diagnostic samples, and this might be one contributing factor to the dampened IL-10 signaling potential of the blasts (Figure 2f and Figure 6c,d). Similarly, the reduced proportion of blasts expressing surface CD40 coupled with the markedly reduced level of CD40 expression (CD40 MFI) could explain the attenuated phosphorylation of p65-NFkB in the blasts treated with trimerized CD40L compared to mature B cells in the diagnostic samples (Figure 2g and Figure 6e,f).

There is a plethora of published evidence highlighting the role of IFNγ in immunosurveillance, but there also exists an expanding body of work attributing a pro-tumorigenic and immune-editing role for IFNγ [53,67,82,83,84,85,86,87,88,89]. For instance, to minimize inflammation-induced collateral tissue damage, IFNγ signaling in normal cells can upregulate immune checkpoint molecules such as PD-L1, PD-L2 and CTLA-4 [67,89,90,91,92,93,94]. Signaling mediated by PD-L1, PD-L2 and CTLA-4 functions as a molecular brake that dampens T cell responses and can also orchestrate the functional exhaustion of T cells [95,96]. Cancer cells have learned to co-opt this protective mechanism as a means of counteracting immunosurveillance, as they can upregulate PD-L1 in response to IFNγ exposure [67,89,90,91,92,93,94,97,98]. Along similar lines, IFNγ signaling can also boost the expression of non-classical MHC class Ia genes, such as HLA-G, -E and -F, which are known to endow cancer cells with ability to evade CTL and NK cell responses [67,85]. Furthermore, IFNγ signaling is also involved in the trafficking of myeloid-derived suppressor cells into the TME [67,88]. Data from IFNγ transgenic mouse models have also underscored the role of elevated IFNγ levels in arresting B cell maturation at the pro-B cell stage [99]. Hence, in light of these observations, it is tempting to speculate that, with reference to our data demonstrating the preservation of IFNγ signaling in B-ALL blasts, this signaling pathway in leukemic B-ALL blasts might serve to facilitate immune escape and promote the survival of these cells (Figure 2e).

STAT5 signaling exerts considerable influence over B cell ontogeny, as evidenced by the fact that its abrogation arrests B cell maturation at the progenitor stage [100,101]. Appropriate checks and balances for STAT5 signaling are essential for preserving the integrity of the B cell developmental pathway because constitutive STAT5 activation is associated with the development of several myeloid and lymphoid malignancies [100,102]. It has also been reported that constitutively active STAT5b can collude with aberrant pre-B cell receptor (pre-BCR) signaling entities to prompt the development of B-ALL [100,103,104,105]. Interestingly, even in the absence of IL-7Rα mutations, the over-expression of wild-type IL-7Rα can itself be oncogenic, given that IL-7 signaling promotes survival (via the JAK3-STAT5 pathway) as well as proliferation (via the PI3Kinase-mTOR pathway) [58,105,106]. In our study, treatment with exogenous IL-7 significantly augmented STAT5 phosphorylation in leukemic blasts compared to both mature B cells in leukemic samples and hematogones in non-leukemic bone marrow specimens (Figure 2a and Appendix A). It is possible that IL-7Rα variants could be contributing to the enhanced IL-7Rα-mediated STAT5 phosphorylation in the leukemic blasts; hence, molecular analyses of the *IL7R* gene will be the focus of a follow-up study [56,105]. Consistent with the observation of enhanced IL-7 Rα-mediated STAT5 phosphorylation in the leukemic blasts, a significantly greater proportion of leukemic blasts expressed IL-7Rα compared to mature B cells in the diagnostic samples, as well as hematogones in non-leukemic bone marrow samples (Figure 3a,c). In addition, in the leukemic samples, the density of surface IL-7Rα expression on the blasts was significantly higher compared to the mature B cells (Figure 3b). Overall, these data and the aforementioned basal hyperphosphorylation of STAT5 align well with reports in the literature that implicate an oncogenic role for IL-7Rα over-expression and dysregulated STAT5 activation in B-ALL.

We also examined STAT3 activation in the context of IL-10 and IL-21 signaling. The leukemic blasts displayed significantly elevated levels of STAT3 basal phosphorylation compared with mature B cells (in diagnostic B-ALL samples) and hematogones (in non-leukemic bone marrow samples) (Figure 1b; Appendix A). However, STAT3 phosphorylation in response to IL-21 and IL-10 treatment was significantly diminished in the leukemic blasts compared to the mature B cells (Figure 2d,f). There is some evidence in the literature linking IL-10 deficiency with dysbiosis, which can in turn promote B cell DNA damage and corrupt B cell development, thus predisposing the host to pediatric B-ALL [36,107], and IL-21 has been shown to promote apoptotic activity in B cells [108]. Both the frequency of cells expressing IL-21R and the density of its expression were significantly lower for the leukemic blasts compared to the mature B cells (Figure 4d; Appendix A). Given these findings, it is tempting to speculate that primitive B cells that transform into leukemic blasts in B-ALL could exploit reduced levels of IL-21R expression to their advantage to avoid IL-21-induced programmed cell death, which might in turn contribute to the proliferative advantage that leukemic cells display. Given the putative link between infectious insults and pediatric B-ALL development, it is worth noting that many pathogens, pathogen products and stressors can engage STAT3 [109,110,111,112,113,114,115,116,117]. Furthermore, persistently activated STAT3 can promote a pro-oncogenic inflammatory environment and subvert anti-tumor immunity [109,118,119,120,121,122,123,124,125,126,127,128]. STAT3 can also cooperate with NFκB to enhance the stemness of cancer cells, and it can also trap the RELA (p. 65) subunit of NFκB in the nucleus, thereby regulating the expression of oncogenic and inflammatory genes that include inflammatory mediators such as IL-6, which can facilitate the establishment of a “feed forward” loop of STAT3 activation [109,118,119,121,122,123,124,129,130,131,132,133,134,135,136,137,138].

It is believed that a dysregulated immune response to pathogens heightens the risk for the development of childhood B-ALL [26,30,31,32,33,34,35]. In this report, we studied the signaling responses to key immunomodulatory mediators to define the phosphoprotein signature in diagnostic samples obtained from pediatric B-ALL patients. The responses measured in leukemic blasts were directly compared against mature B cells and T cells residing in the same samples. Our data reveal evidence of generalized basal hyperphosphorylation restricted to the blast population, and exposure to immunomodulatory mediators displayed evidence of both symmetry and asymmetry in the phosphoprotein signature between the blasts and mature B cells. This indicates the existence of persistent inflammatory stimuli within the TME, and the similarities and disparities in the immunomodulatory-induced signaling patterns likely reflect differences in cognate receptor expression patterns between blasts and mature B cells in the diagnostic pediatric B-ALL samples. Additionally, the phosphoprotein signature detected in the B-ALL blasts in this study is also quite distinct from that detected in lymphocytes that we previously evaluated in healthy control subjects, as well as in perinatally infected pediatric subjects with Human Immunodeficiency virus (HIV) infection and early anti-retroviral therapy (ART) initiation but displaying disparate patterns of viral control and attendant underlying inflammation [48]. Finally, another notable aspect of our study is the evidence of significantly elevated basal phosphorylation in the leukemic blasts present in the diagnostic samples of the study subjects that did not survive the disease compared with those that did (Figure 7). In future studies, we will further explore this interesting observation as a potential prognostic indicator of the disease.

## 4. Materials and Methods

### 4.1. Study Subjects

We are a Clinical Laboratory Improvement Amendments (CLIA)-certified and College of American Pathologists (CAP)-accredited diagnostic immunology and flow cytometry laboratory. Sixty-three pediatric subjects were recruited into this study after a confirmed diagnosis of B-acute lymphoblastic leukemia (B-ALL) rendered by a board-certified hematopathologist following a flow cytometric evaluation of peripheral blood (*n* = 53) or bone marrow (*n* = 10) samples in our laboratory (Appendix A). Diagnostic samples from each study subject were also independently evaluated for chromosomal aberrations (including but not limited to hyper-diploidy, chromosomal translocations, fusions, rearrangements and loss) by a board-certified cytogeneticist. For assays requiring assessments of hematogones, left-over samples from de-identified leukemia-negative bone marrow specimens were analyzed. This study was carried out in accordance with the recommendations of the Human Research Protection Program Guidelines of our Institutional Review Board (IRB). This protocol was approved by our IRB (No. 2014-15677; Study Title: To Define a Cellular Signature That Predicts Disease Relapse in Precursor-B cell Acute Lymphocytic Leukemia in Pediatric Patients; and No. 2020-3756: Validation of new screening tests to aid in the diagnosis of hematologic malignancies (leukemias and lymphomas)). All B-ALL patients ≥ 18 years of age gave written informed consent in accordance with the Declaration of Helsinki. B-ALL patients between the ages of 12 and 17 years provided written assent, and in addition, we obtained written informed consent from their legal guardians. Written informed consent was obtained from the legal guardians of all B-ALL patients under the age of 12 years.

### 4.2. Phosflow

We performed phosflow staining using established protocols [48,50,139,140]. Please refer to the gating strategy in Appendix A [141,142,143,144,145]. Briefly, aliquots from peripheral blood or bone marrow diagnostic samples were separately treated for 15–20 min at 37 °C with either phosphate-buffered saline (PBS) or the following stimulants: recombinant human (rh) IFNγ, rhIL-4, rhIL-10 (BD Biosciences, San Jose, CA, USA), rhIL-2 (PeproTech, Rocky Hill, NJ, USA), rhIL-7 (Thermo Fisher Scientific, San Diego, CA, USA), rhIL-21 (Biolegend, San Diego, CA, USA) and trimerized rhCD40L (Enzo Life Sciences, Inc. Farmingdale, NY, USA), within 24 h following sample collection. The cells were then fixed, permeabilized and stained with monoclonal antibodies (MAb) targeting the following molecules: anti-human cytoplasmic CD3 (cyCD3; clone UCHT1) and CD10 (clone HI10a), CD34 (clone 8G12), anti-human phospho-STAT1 (pY701; clone 4a), phospho-STAT3 (pY705; clone 4/P-STAT3), phospho-STAT5 [clone 47/Stat5(pY694)], phospho-STAT6 (pY641; clone 18/P-Stat6), anti-NFκB p65 (pS529; clone K10-895.12.50) (all from BD Biosciences, San Jose, CA, USA), anti-human cyCD79a (clone HM57;Agilent [Dako]; Santa Clara, CA, USA) and anti-human CD45 (clone HI30; ThermoFisher Scientific, Waltham, MA, USA). Our preliminary analyses revealed a preponderant lack of statistically significant differences in the phosphorylation signal measured for the blasts and mature B cells between the peripheral blood and bone marrow samples). Hence, the phosphorylation signal values for the peripheral blood and bone marrow sample donors were not further segregated based on sample type for this study. A total of 50,000–100,000 events/tube were acquired on FACS Canto-II instruments (Becton-Dickinson, Franklin Lakes, NJ, USA), and the data were analyzed using Cytobank software (version ≤ 10.3) (Cytobank (now part of Beckman Coulter) Santa Clara, CA, USA). Each study subject was analyzed once. The phosflow data for the leukemic blasts and mature B cells and T cells, evaluated for the specimens obtained from each study subject and based on the clinical subtypes used for the stratification of BCP-ALL patients, are depicted in Appendix A.

### 4.3. Cell Surface Receptor Expression

Flow cytometry was utilized to examine the cell surface expression of IL-2Rα (CD25), IL-4Rα (CD124), IL-7Rα (CD127), IL-10Rα (CD210), IL-10Rβ, IL-21R (CD360), IFNγR1 (CD119), IFNγR2, common γ chain (CD132) and CD40 on T cells, non-leukemic B cells and leukemic blasts in peripheral blood or bone marrow samples within 24 h of collection (please refer to the gating strategy in Appendix A) [141]. Hematogones were evaluated in de-identified leukemia-negative bone marrow samples within 24 h of collection (please refer to the gating strategy in Appendix A) [142,143,144,145]. The following Ab reagents were utilized for this assessment: (a) From Biolegend (San Diego, CA, USA): anti-human IL-7Rα (clone A019D5), anti-human IL-10Rα (clone 3F9), anti-human IFNγR1 (clone GIR-94), anti-human IL-2Rα (clone BC96), anti-human IL-4Rα (clone G077F6), anti-human IL-21R (clone 2G1-K12), anti-human CD40 (clone 5C3), anti-human common γ chain (clone TUGh4), mouse IgG1,κ (clone MOPC-21), rat IgG2a,κ (clone RTK2758), mouse IgG2b,κ (clone MPC-11), mouse IgG2a,κ (clone MOPC-173) and rat IgG2b,κ (clone RTK4530). (b) From BD Biosciences (San Jose, CA, USA): anti-human IL-10Rβ (clone 90220) and mIgG1,κ (clone X40). (c) From R&D Systems (Minneapolis, MN, USA): anti-human IFNγR2 (polyclonal goat IgG) and polyclonal goat IgG isotype control. The isotype control Abs were used at matching doses to the Abs targeting the indicated cell surface receptors, and both the isotype control and cell-surface-receptor-directed Abs were cross-adsorbed with normal mouse, rat and goat serum (2% *v*/*v*), as appropriate, prior to use in the assays. T cells, non-leukemic B cells, leukemic blasts and hematogones were identified via Abs targeting CD45, CD3, CD19 and CD10 (or CD34 for samples lacking CD10 expression). A total of 250,000 events/tube were acquired on FACS Canto-II instruments (Becton-Dickinson, Franklin Lakes, NJ, USA), and the data were analyzed using FlowJo^TM^ v10.7 (or higher) software (BD Life Sciences). Each study subject was analyzed once.

### 4.4. Statistical Analyses

In the experiments where we compared data from more than two groups, the D’Agostino–Pearson omnibus normality test (*α*  =  0.05) for normal distributions was performed followed by the Kruskal–Wallis and Dunn’s multiple comparisons tests (*α*  =  0.05) to identify any potential statistically significant differences in the patterns measured between the groups (Prism version ≥ 8.0, GraphPad, San Diego, CA, USA). In some of the experiments where we compared more than two groups, if the numeric values being compared represented matched data, the Friedman and Dunn’s multiple comparisons tests (*α*  =  0.05) were utilized. When we compared data from two groups, the test for normality was run as described above, following which we employed either the unpaired two-tailed Student’s *t* test (with Welch’s correction) or the two-tailed Mann–Whitney *U* test, as appropriate (Prism version ≥ 8.0, GraphPad, San Diego, CA, USA). For all statistical analyses, measurements were taken from distinct samples. The figure legends indicate the number of times each sample was evaluated.

## Figures and Tables

**Figure 1 ijms-24-13937-f001:**
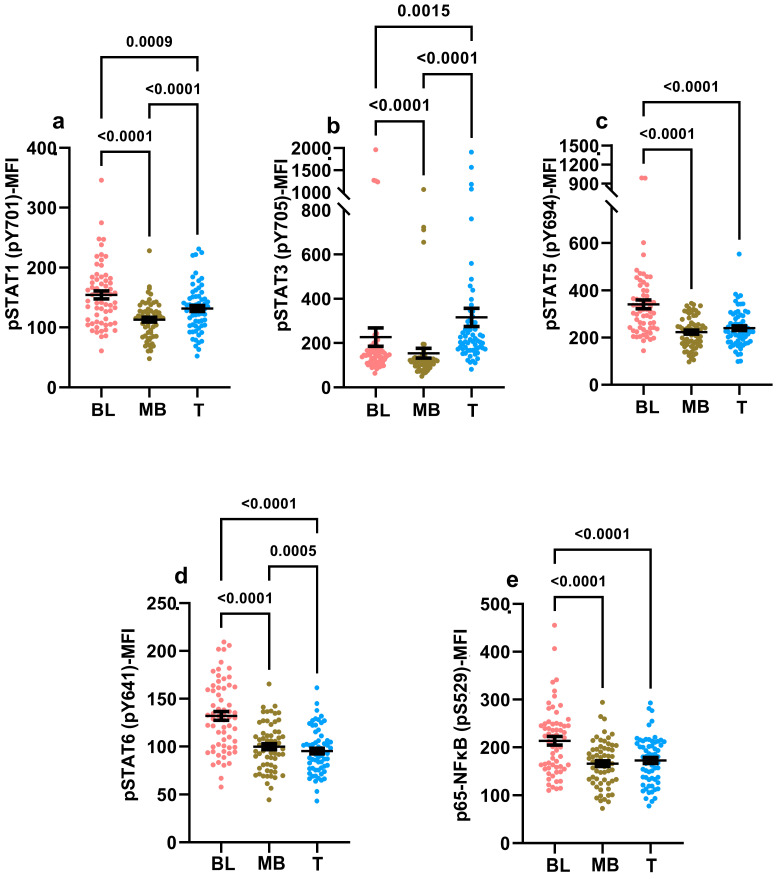
Basal hyperphosphorylation of proximal signaling nodes. The basal phosphorylation signal for the signaling nodes listed along the y-axes was measured following PBS treatment via phosflow (**a**–**e**). The phosphorylation signal data are depicted as MFI for the blasts (BL; salmon-colored filled circles), normal mature B cells (MB; green-colored filled circles) and T cells (T; blue-colored filled circles) identified in the diagnostic samples of confirmed cases of pediatric B-ALL (*n* = 63). Statistically significant differences were identified based on the statistical tests applied, as listed in the Materials and Methods section. Individual data points are shown in addition to the mean ± standard error of mean (SEM). A *p*-value < 0.05 was considered significant. Each study subject was evaluated once.

**Figure 2 ijms-24-13937-f002:**
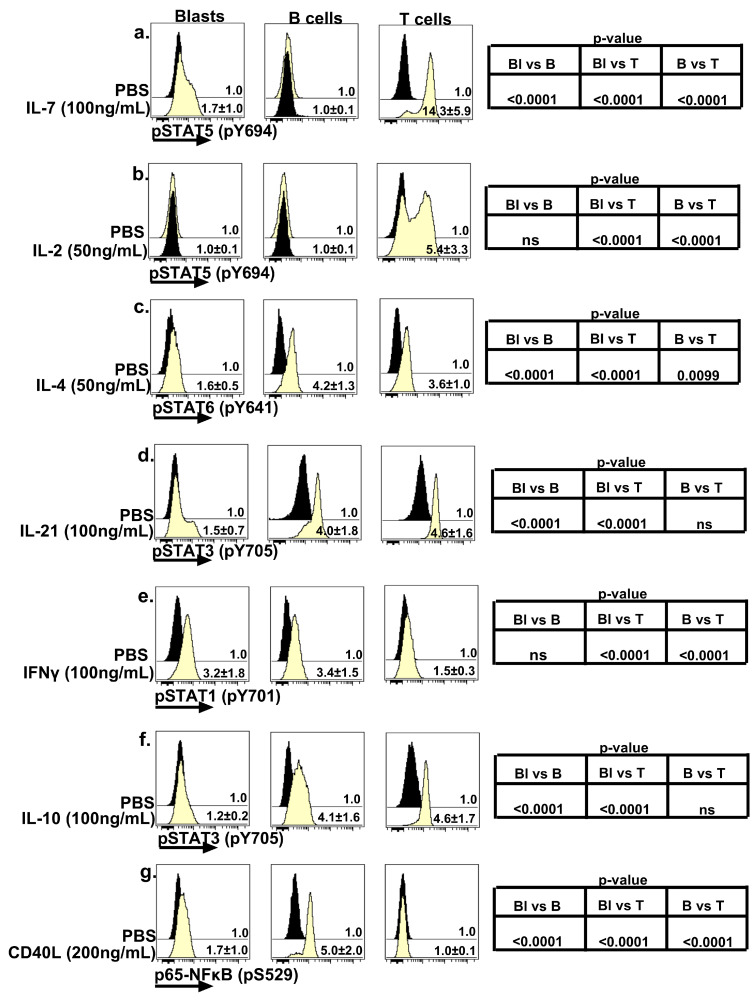
Immunomodulatory mediator-induced phosphorylation of proximal signaling nodes in pediatric B-ALL diagnostic samples. Representative data depicting the phosphorylation signal for the signaling nodes listed along the x-axes were measured via phosflow following treatment with the indicated immunomodulatory mediators (rhIL-7; rhIL-2; rhIL-4; rhIL-21; rhIFNγ; rhIL-10, trimerized CD40L) (**a**–**g**). The phosphorylation signal data are depicted in the form of stacked histogram overlaid plots. The numerical values in each stacked histogram plot denote the mean ± standard deviation (SD) of the fold change in the MFI of the phosphorylation signal of the signaling node following treatment with the signaling input compared to PBS treatment, which is normalized to one. Statistically significant differences were identified based on the statistical tests applied as listed in the Materials and Methods section. A *p*-value < 0.05 was considered statistically significant (ns: not significant). Each study subject was evaluated once. *n* = 63 for IFNγ, IL-2, IL-4, IL-10 and CD40L; *n* = 59 for IL-7; *n* = 43 for IL-21. Bl: Blasts; B: Mature B cells; T: Mature T cells.

**Figure 3 ijms-24-13937-f003:**
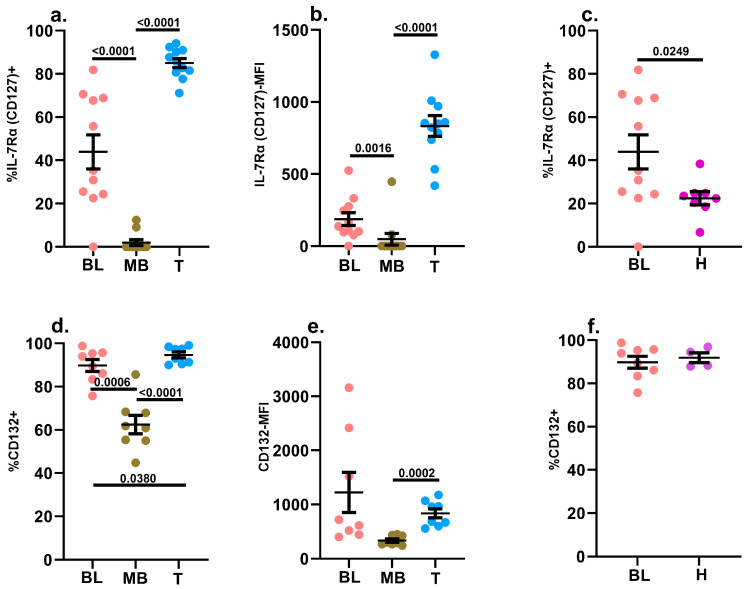
Assessment of IL-7Rα (CD127) and common γ chain (CD132) surface expression. The background-subtracted frequencies (%) of blasts (BL), mature B cells (MB) and mature T cells (T) in diagnostic samples of pediatric B-ALL patients (*n* = 11) expressing CD127 (**a**) and its signaling subunit (CD132) (**d**) on their surfaces were assessed via flow cytometry. The background-subtracted MFI of CD127 (**b**) and CD132 (**e**) surface expression was also assessed. The background signal was determined by staining an aliquot of each specimen with dose-matched isotype control antibodies. Individual data points are shown in addition to the mean ± SEM. The background-subtracted frequency (%) of CD127 (**c**) and CD132 (**f**) expressing cells was also compared between the blasts present in leukemic samples, and hematogones present in non-leukemic bone marrow specimens (*n* = 4–8). A *p*-value < 0.05 was considered significant. Each study subject was evaluated once.

**Figure 4 ijms-24-13937-f004:**
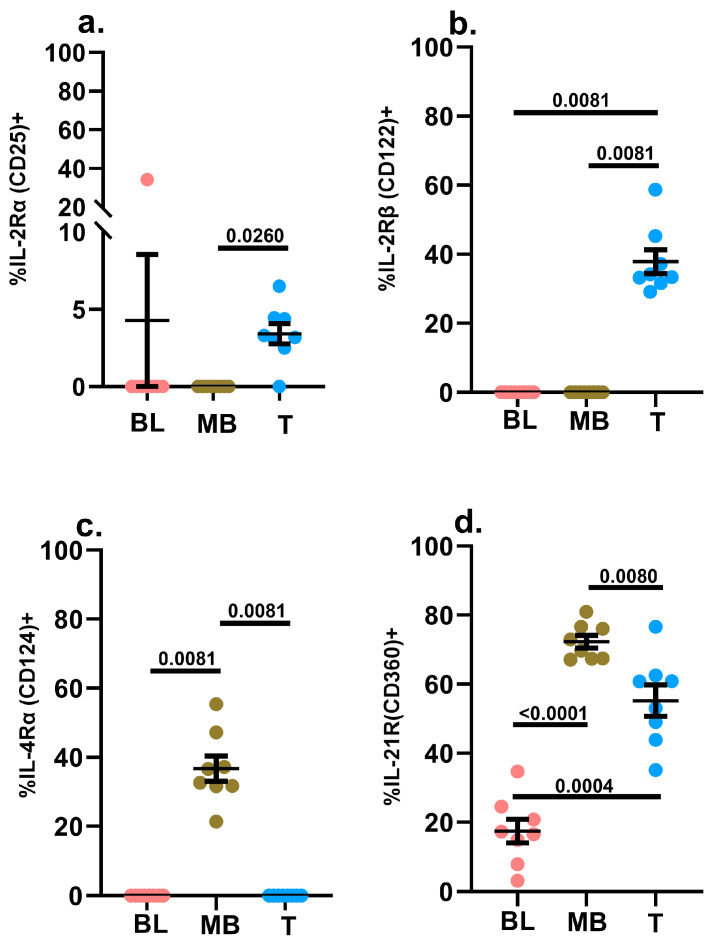
Evaluation of IL-2R, IL-4R and IL-21R surface expression. The background-subtracted frequencies (%) of blasts (BL), mature B cells (MB) and mature T cells (T) in diagnostic samples of pediatric B-ALL patients (*n* = 8) expressing IL-2Rα (CD25) (**a**), IL-2Rβ (CD122) (**b**), IL-4Rα (CD124) (**c**) and IL-21R (CD360) (**d**) on their surfaces were assessed via flow cytometry. The background signal was determined by staining an aliquot of each specimen with dose-matched isotype control antibodies. Individual data points are shown in addition to the mean ± SEM. A *p*-value < 0.05 was considered significant. Each study subject was evaluated once.

**Figure 5 ijms-24-13937-f005:**
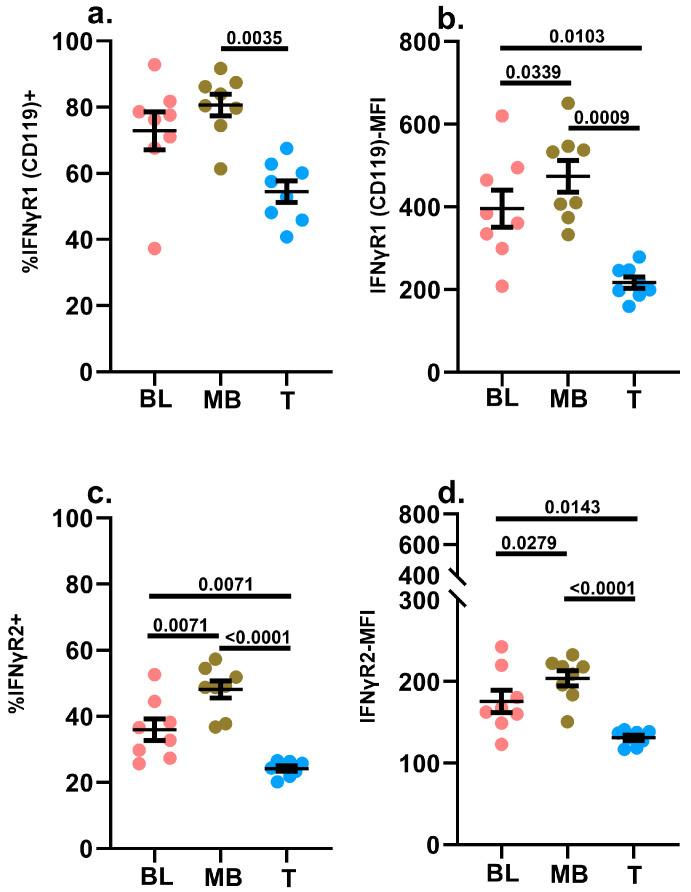
Determination of IFNγR1 (CD119) and IFNγR2 surface expression. The background-subtracted frequencies (%) of blasts (BL), mature B cells (MB) and mature T cells (T) in diagnostic samples of pediatric B-ALL patients (*n* = 8) expressing IFNγR1 (CD119) (**a**) and IFNγR2 (**c**) on their surfaces were assessed via flow cytometry. The background-subtracted MFI of IFNγR1 (**b**) and IFNγR2 (**d**) surface expression was also assessed. The background signal was determined by staining an aliquot of each specimen with dose-matched isotype control antibodies. Individual data points are shown in addition to the mean ± SEM. A *p*-value < 0.05 was considered significant. Each study subject was evaluated once.

**Figure 6 ijms-24-13937-f006:**
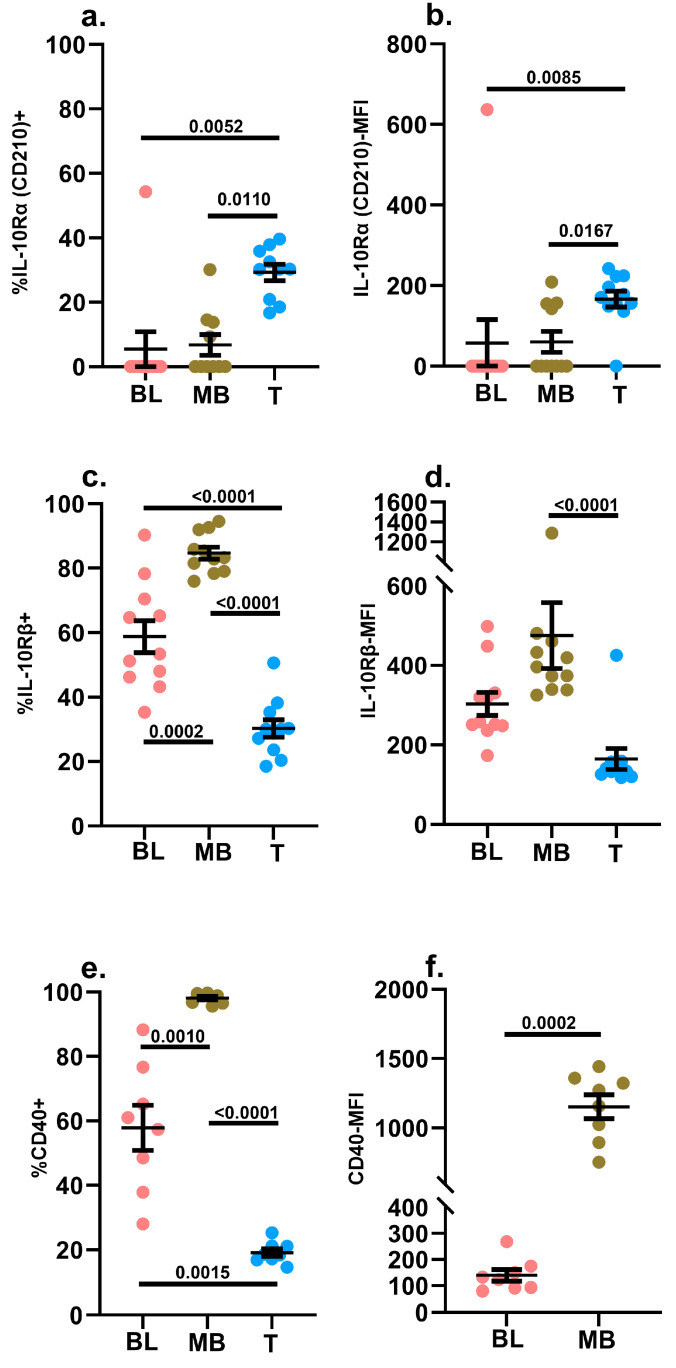
Measurement of IL-10Rα (CD210), IL-10Rβ and CD40 surface expression. The background-subtracted frequencies (%) of blasts (BL), mature B cells (MB) and mature T cells (T) in diagnostic samples of pediatric B-ALL patients (*n* = 8–11) expressing IL-10Rα (CD210) (**a**), IL-10Rβ (**c**) and CD40 (**e**) on their surfaces were assessed via flow cytometry. The background-subtracted MFI of IL-10Rα (CD210) (**b**), IL-10Rβ (**d**) and CD40 (**f**) surface expression was also assessed. The background signal was determined by staining an aliquot of each specimen with dose-matched isotype control antibodies. Individual data points are shown in addition to the mean ± SEM. A *p*-value < 0.05 was considered significant. Each study subject was evaluated once.

**Figure 7 ijms-24-13937-f007:**
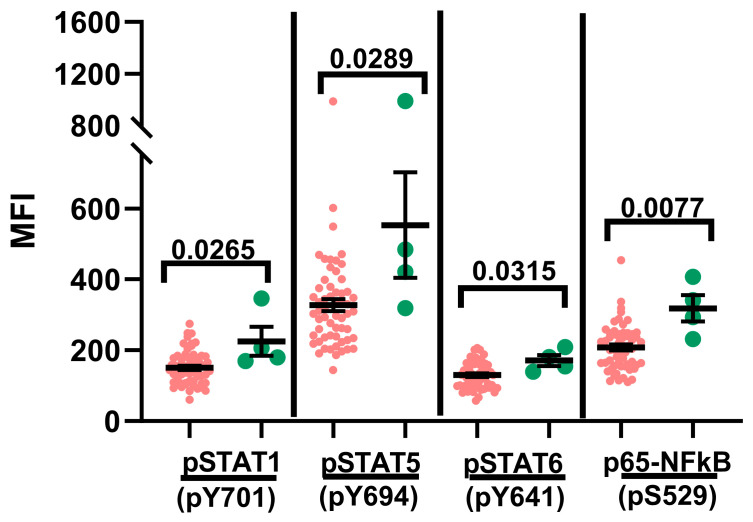
Pattern of basal hyperphosphorylation of select proximal signaling nodes in diagnostic samples based on overall survival (OS) analyses. The basal phosphorylation signal for the signaling nodes listed along the x-axes was measured following PBS treatment via phosflow. The phosphorylation signal data are depicted in the form of median fluorescence intensity (MFI) for the leukemic blasts in patients that either survived (pink circles; *n* = 58) or died (green circles; *n* = 4). A *p*-value < 0.05 was considered significant. Each study subject was evaluated once.

## Data Availability

The data presented in this study are available upon reasonable request from the corresponding author.

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
