# Peer review of "Defining the Basal and Immunomodulatory Mediator-Induced Phosphoprotein Signature in Pediatric B Cell Acute Lymphoblastic Leukemia (B-ALL) Diagnostic Samples"

_ijms, 2023, doi:10.3390/ijms241813937_

Round 1

Reviewer 1 Report

Khanolkar and colleagues started with the hypothesis that dysregulated immune responses to infectious insults contribute to the development of pediatric B-ALL. However, this hypothesis to me it is not sufficiently addressed by the experiments proposed. The study is very descriptive and no functional experiments are presented. The conclusions, the novelty and the contribution to the field are not clear to me. Authors should sum up the most important concepts highlighting the novelty (if present) and their contribution to the field.

Major:

1.  To further prove their hypothesis authors should maight test the same set of cytokines experiments in a subset of patients with for instance inflammatory disease to prove a different pattern, if present, compared to blast cells. Also healthy donors could be of help.

2.      IL-7 response of blast cells compared to normal one might be due to the presence of mutations in the IL-7R as reported also in the literature. Is that the case?

3. More importantly, BCP-ALL patients are not stratified only on the basis of white blood cell count (not used anymore in clinical routine), and the presence of BCR-ABL or 4;11 translocation. What about 1;19 and 12;21 patients? CRLF2 rearranged/mutated one (in which STAT5 is hyperphosphorylated)? Authors should show different phosphorylation levels based on all the clinical subtypes used for stratification of BCP-ALL patients reported in the supplementary table.

Minor:

Indicate in the graphs the exact phosphorylation residue.

Mandatory to sum up and simplify the introduction and results.

Reviewer 2 Report

The authors analyzed 63 bone marrow samples from pediatric patents with B precursor ALL by phospho flow cytometry. They found increased basal phosphorylation of STAT1 , STAT3. STST5, STAT6 and p65-NKkB in leukemic blasts compared to mature B cells in the same samples. Stimulated responses to Il7 (pSTAT1) were increased while those to Il4 (pSTA6), IL21 (pSTAT3), IL10 (pSTAT3) and CD40L (p65-NFkB) were reduced. Changes of expression levels for cognate receptors were largely consistent with measured stimulated responses. Basal and stimulated phospharylation levels for selected pathways did not corelate with ALL risk group, relapse or survival.       

Comments

1.       The authors provide a good overview in the introduction of hypotheses linking immune responses and inflammation to development of ALL. This section is well written and comprehensive.  However, they do not state and test a hypothesis.

2.       It remains unclear if the increased basal phosphorylation level detected in ALL blasts is a cell-intrinsic phenomenon or induced by the microenvironment. Neither this big question is answered nor the one which stimuli generate this elevated level. A a result the reports remains descriptive.

3.       The observation that phosphorylation responses correlate with receptor expression levels is not entirely surprising and does not advance our understanding markedly.  

4.       Basal phosphorylation level in ALL blasts was not correlated with risk group, elapse and survival in an analysis hampered by sample size. Were there any differences of stimulated phosphorylation levels (e.g. IL7/STAT5)?  Were there differences according to cytogenetic  groups within the 28 SR or 33 HR/VHR ALL samples? If not, conclusions and applicability are limited.

5.       Bulk populations of blasts and B cells are analyzed. The limitation of this approach should be discussed.   

6.       Were there differences between normal B and T cells in the ALL vs. non-leukemic control samples? How were figures 3 c and F constructed? The data points for ALL blasts seem identical to Fig. 3a and d.  

7.       It appears that stimulated responses to IFN gamma in ALL blasts are referred to as different in the discussion (line 432). This is confusing since Fig 2e showed absence of such a difference.  

8.       The disussion is too long, speculative and lacks focus.   

  Minor points

1.       State that samples analyzed were bone marrow samples

2.       Which germline variants are being referred to (line 62)?

3.       Does Fig.2 show representative examples ? If so, pls mention this fact.

4.       Suggest to rephrase the presentation of results and interpretation of Fig. 7 and use the terms  event-free and overall survival rather than decedents and relapse

In sum, increased basal hyperphosphorylation and and Il-7-stimulated STAT5 phosphorylation are reported in a cohort of ALL samples. Conclusions are limited.   

Round 2

Reviewer 1 Report

Khanolkar and colleagues started with the hypothesis that dysregulated immune responses to infectious insults contribute to the development of pediatric B-ALL. However, this hypothesis to me it is not sufficiently addressed by the experiments proposed. The study is very descriptive, and no functional experiments are presented. The conclusions, the novelty and the contribution to the field are not clear to me. Authors should sum up the most important concepts highlighting the novelty (if present) and their contribution to the field.

Major:

1.  To further prove their hypothesis authors should might test the same set of cytokines experiments in a subset of patients with for instance inflammatory disease to prove a different pattern, if present, compared to blast cells. Also healthy donors could be of help.

2.      IL-7 response of blast cells compared to normal one might be due to the presence of mutations in the IL-7R as reported also in the literature. Is that the case?

3. More importantly, BCP-ALL patients are not stratified only on the basis of white blood cell count (not used anymore in clinical routine), and the presence of BCR-ABL or 4;11 translocation. What about 1;19 and 12;21 patients? CRLF2 rearranged/mutated one (in which STAT5 is hyperphosphorylated)? Authors should show different phosphorylation levels based on all the clinical subtypes used for stratification of BCP-ALL patients.

Minor:

Indicate in the graphs the exact phosphorylation residue.

Mandatory to sum up and simplify the introduction and results

Author Response

Please see the attached pdf file for a point-by-point response to the reviewer comments.  Thank you.

Reviewer 2 Report

The authors have made a satisfactory effort to address the points raised.  

Author Response

(The authors gave the same response as above.)

Round 3

Reviewer 1 Report

Authors adressed the questions asked.